

# Random and externally controlled occurrence of Dansgaard-Oeschger events

Johannes Lohmann[1] and Peter D. Ditlevsen[1]

[1]Centre for Ice and Climate, Niels Bohr Institute, University of Copenhagen, Denmark

**Correspondence:** Johannes Lohmann (johannes.lohmann@nbi.ku.dk)

**Abstract.** Dansgaard-Oeschger (DO) events constitute the most pronounced mode of centennial to millennial climate variability of the last glacial period. Since their discovery, many decades of research have been devoted to understand the origin and nature of these rapid climate shifts. In recent years, a number of studies have appeared that report emergence of DO-type variability in fully coupled general circulation models via different mechanisms. These mechanisms result in the occurrence of DO events at varying degrees of regularity, ranging from periodic to random. When examining the full sequence of DO events as captured in the NGRIP ice core record, one can observe high irregularity in the timing of individual events at any stage within the last glacial period. In addition to the prevailing irregularity, certain properties of the DO event sequence, such as the average event frequency or the relative distribution of cold versus warm periods, appear to be changing throughout the glacial. By using statistical hypothesis tests on simple event models, we investigate whether the observed event sequence may have been generated by stationary random processes or rather has been strongly modulated by external factors. We find that the sequence of DO warming events is consistent with a stationary random process, whereas dividing the event sequence into warming and cooling events leads to inconsistency with two independent event processes. As we include external forcing, we find a particularly good fit to the observed DO sequence in a model where the average residence time in warm periods are controlled by global ice volume and cold periods by boreal summer insolation.

## 1 Introduction

During the last glacial period, lasting from approximately 120 kyr BP to 12 kyr BP (kiloyears before present), a large number of abrupt large-scale climate changes have been recorded in Greenland ice cores and other Northern Hemisphere climate proxies. These so-called Dansgaard-Oeschger (DO) events (Dansgaard et al., 1993) are characterized by an abrupt warming of 10-15 K from cold conditions (stadials) to warmer conditions (interstadials) within a few decades. This is typically followed by gradual cooling, lasting centuries to thousands of years, until a more abrupt jump back to cold conditions is observed. The warming events are not regularly spaced over the glacial, but rather distributed in a complex temporal pattern, as can be seen in the NGRIP ice core record in Fig. 1. This raises questions about the causes of these recurring climate changes. Could an internal oscillation of large components of the climate system under strongly varying conditions give rise to this pattern? Are the climate changes in contrast manifestations of highly sensitive, multistable climate system components, where jumps in





between different states are triggered in an unpredictable way by one or possibly many different other chaotic components?

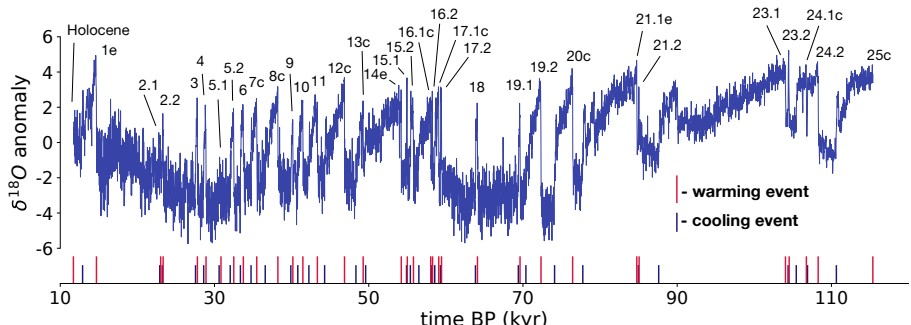

**Figure 1.** NGRIP oxygen isotope ice core record in 20 year binned resolution and associated Dansgaard-Oeschger warming and cooling events. The numbers above the time series indicate the warming transitions into the respective Greenland interstadials. The nomenclature is adopted from (Rasmussen et al., 2014) and only events considered in this study are marked. On the time axis we marked the timing of warming (red) and cooling (blue) events.

Since the discovery of these unexpected climate events with no known cause, questions of this kind have been addressed. Whereas high-resolution coupled climate models under glacial conditions typically lack DO-type variability, models of inter-

mediate complexity and simpler conceptual models have been proposed to explain qualitative features of the sequence of last glacial climate changes. Starting from the discovery of an approximate 1500 year spectral signature in the GISP2 ice core record (Grootes and Stuiver, 1997) and an apparent in-phase pacing of individual events by multiples of this time period (Alley et al., 2001; Schulz, 2002; Rahmstorf, 2003), a number of competing hypotheses have been compared to the data. Among these were studies aiming to establish a mechanism for this periodicity, including direct triggering by periodic forcing (Braun et al.,

2005), stochastic resonance (Alley et al., 2001), ghost resonance (Braun et al., 2007) and coherence resonance (Timmermann et al., 2003). On the other hand, it has been shown that there is limited significance to the periodic spectral signature (Braun et al., 2010) and pacing of single events (Ditlevsen et al., 2007). When including data reaching further back in time than 50 ky BP it is found that only very weak periodic contributions to modeled switching sequences are compatible with the data and that instead it is more likely that the observed sequence of events is a realization of a purely noise-driven process (Ditlevsen

et al., 2005).

In this work, we want to expand on this idea by testing whether the observed sequence of events is indeed consistent with one or more random, stationary processes, or whether the changes over time of the properties of the observed event sequence require a modulation of parameters of the governing process over time. To this end, we regard the whole glacial period, as

opposed to previous efforts focusing on a rather regular period in the middle of the glacial. We investigate two different levels in detail of description by first only regarding the sequence of warming events and second the combined sequence of alternating transitions in between cold and warm conditions. We proceed by testing two null hypotheses: 1. The sequence of DO



warming events is a realization of a Poisson process with fixed rate parameter. 2. The sequence of stadials and interstadials is a realization of two independent Poisson processes with fixed rate parameters giving rise to transitions in between stadials and interstadials. In order to test the hypotheses, we regard the evolution of the number of warming events in a moving window of 20 ky. This quantity strongly deviates from a constant occurrence of events over time in the DO sequence. We test whether

samples from the above mentioned stationary processes show a similar irregularity over time.

In addition to the evolution of the frequency of warming events we look at the evolution of the abundance of the stadial over the interstadial condition, which changes significantly over time in the DO sequence. This additional non-stationary structure in the data is the basis for another hypothesis test we perform. Finally, we test how the models' support with respect to the

data is improved as we force the rate parameters with a combination of a global climate proxy and orbital variations of insolation, to incorporate changing background climate conditions. The main findings of this study are: 1. A Poisson process with fixed rate parameter, modeling warming transitions only, is consistent with the time variations in the NGRIP DO warming event sequence. 2. A model composed of two independent stationary Poisson processes governing transitions in between stadials and interstadials is not consistent with the time variations in the observed DO event sequence. 3. Forcing the afore-

mentioned models with a combination of a global ice volume proxy and a summer insolation curve leads to good statistical agreement with the observed sequence. Specifically, we find good agreement for a model with two individual processes, where the average transition rate from interstadial to stadial is controlled by global ice volume forcing, obtained from independent ocean core isotope records, and the average transition rate from stadial to interstadial is controlled by boreal summer insolation.

The paper is structured in the following way. In Section 2 we introduce in more detail the data used in this study, the summary statistics used to investigate irregularity, the models used to explain the data and the hypothesis test procedure. In Section 3 we present the results of the hypothesis tests on the different models. We discuss and interpret the results in Section 4.

## 2    Methods and Models

Our study of the sequence of DO events is based on the refined dating represented by the GICC05 time scale Svensson et al.

(2006), the classification of Greenland stadials (GS) and Greenland interstadials (GI) given in (Rasmussen et al., 2014) and the timings reported therein. We consider all stadials and interstadials and corresponding transitions, starting with GI-25c at 115370 kyr BP and ending with the transition from GS-1 to the Holocene at 11703 kyr BP. We do not include events classified as subevents, i.e., drops in the middle interstadials to colder, but not fully stadial conditions, with the exception of GS-14. This yields a total number of 34 warming events and 33 cooling events. This increase in number from the 25 originally reported

warming events is due to refined subdivision (Rasmussen et al., 2014).

Given sequence and timing of transitions in between stadials and interstadials, we construct time-varying indicators of non-stationarity in the sequence of events, which are shown in Fig. 2a,b. To this end, we calculate the number of warming transitions





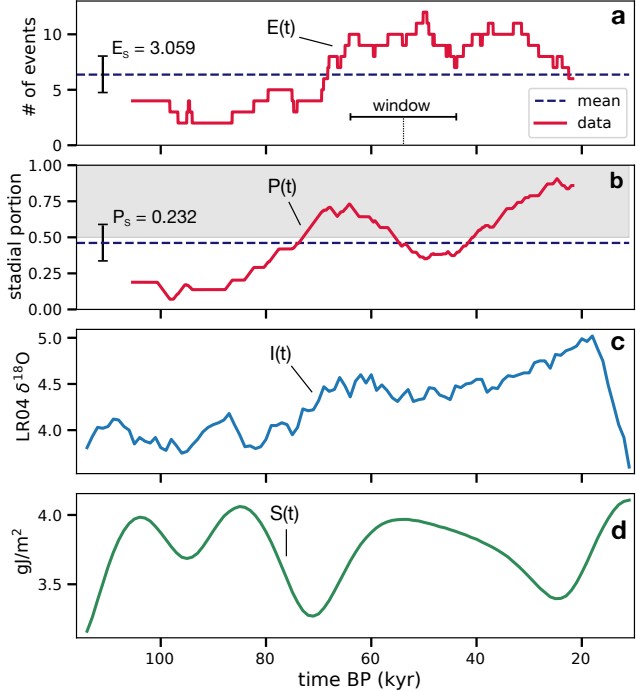

**Figure 2.** Time-varying non-stationarity indicators calculated from the NGRIP DO sequence, and climate forcings. (a): The number of warming events in a running 20 kyr window $E(t)$ (red) and the mean value, corresponding to a regular sequence of events (dashed blue). (b): The abundance of stadials in a 20 kyr window $P(t)$. For values greater than 0.5 (indicated by gray shading) the portion of stadials is larger than the portion of interstadials within the window. (c) Ocean sediment proxy record for global ice volume $I(t)$. (d) Integrated summer insolation at 65 deg North $S(t)$.

withing a moving window of 20 kyr, which we denote as $E(t)$. The window size of 20 kyr is chosen as trade-off in between resolution and statistical robustness of longer-term features in the event sequence given the characteristic time scale of event occurrence of 3.1 kyr. The window is furthermore of comparable size to dominant variations in global background climate and insolation forcing, which will be investigated below. We yield a timeseries indicating the deviation in the occurrence frequency

5  of warming events from a fully regular event occurrence. We summarize this in a scalar test statistic $E_S$ defined as the root mean squared deviation of the timeseries $E(t)$ from the constant $\bar{E} = 6.367$, which the average number of events per 20 kyr of the whole DO sequence. For periodically occurring events, with a period significantly smaller that the window size, we would expect the test statistic $E_S$ to be close to zero. For completely randomly occurring events the test statistic will show a finite value, which depends on the moving window size relative to the expected waiting time in between events. The same

10  time-varying indicator has previously been used to complement a model comparison study aiming to quantify the influence of external forcing to conceptual models of the NGRIP ice core record (Mitsui and Crucifix, 2017). With this statistic we test whether the observed DO sequence departs significantly further from regularity as compared to what is expected by a random,





uncorrelated event sequence. If this is true, it would point to non-stationarity of the underlying process.

While no significant correlation between duration of individual stadials and preceding or subsequent interstadial is observed (Pearson's $r = 0.04$ and $r = -0.15$, respectively), the data suggests long-term variations in stadial and interstadial duration

distributions. If these variations are systematic for stadials and interstadials (i.e. correlated or anti-correlated) they should be detectable in the correlation of individual neighboring stadial/interstadial durations given a large enough sample size. However, due to the small sample size of events in this study and the broad distribution of event waiting times a correlation due to long-term trends is practically not observed. It is thus necessary to devise another time-varying indicator in order to capture additional detail in the structure of the DO sequence. When observing a given number of events in a time window, this may

be either comprised of a combination of long stadials and short interstadials, or short stadials and long interstadials. This is not resolved in the statistic $E_S$. To capture this structure, we investigate the total portion of stadials within a moving window. Given the sum of the duration of all stadials in a time window $T_{st}$, the indicator is defined as $P(t) = T_{st} \cdot (20 \text{ kyr})^{-1}$. We summarize this indicator by a scalar test statistic $P_S$. It is defined as the root mean squared deviation from the expected value $\bar{P} = 0.461$, which is the sum of all stadial durations divided by the total length of the record.


We now describe the models which are used to evaluate our hypotheses on the data using the test statistics described above. The first model used in our study models the process generating the sequence of warming events as a Poisson process with fixed rate parameter $\lambda$, i.e., we disregard the cooling transitions in between warming events. It is denoted as 'one-process model' hereafter. The inverse of the rate parameter corresponds to the average waiting time in between warming events. The Poisson

process corresponds to a situation where there is no memory of the past and thus the probability for a transition is determined by $\lambda$ and is independent of time. All information on climate stability is represented in the parameter $\lambda$. We set its value equal to the inverse of the empirically observed average waiting time in the full glacial record. This yields $\lambda = (3.141 \text{ kyr})^{-1}$.

As a second model, labeled 'two-process model' hereafter, we propose two individual processes for generating warming

transitions from stadials to interstadials and cooling transitions from interstadials back to stadials. Each is represented by a Poisson process with fixed rates $\lambda_1$ and $\lambda_2$, for warming and cooling, respectively. Again, the parameters are derived from the data by considering the empirical average residence times in stadials and interstadials, yielding $\lambda_1 = (1.477 \text{ kyr})^{-1}$ and $\lambda_2 = (1.663 \text{ kyr})^{-1}$. The model is different from the one previously introduced in that the sequence of warming transitions will not be a Poisson process, but a more regular one that is obtained from the sum of two independent processes. The probability

distribution of waiting times $T$ in between warming events is not exponential, but can be evaluated, yielding

$$P(t > T) = (\lambda_1 - \lambda_2)^{-1} \cdot (\lambda_1 e^{-\lambda_1 T} - \lambda_2 e^{-\lambda_1 T}). \tag{1}$$

The average interstadial and stadial durations of the data seem to behave differently over the course of the glacial, as captured by our second test statistic. This motivates us to study whether this behavior is likely to be encountered by chance assuming





randomness and independence of both up- and down-switches.

As comparison to our hypothesis of stationary random processes, we consider the same models with time-varying rate parameters, which are given by a linear combination of two external climate factors: $\lambda = \lambda_0 + a_1 S(t) + a_2 I(t)$. Firstly, a measure of incoming solar radiation at 65 deg North integrated over the summer $S(t)$ (Huybers, 2006). It is defined as the annual sum of the insolation on days exceeding an average of 350 W/m$^2$. Secondly, the LR04 ocean sediment record stack as proxy for global ice volume $I(t)$ (Lisiecki and Raymo, 2005). We note that, in contrast to insolation, global ice volume is not an external factor in the strict sense. However, it's dominant variability is on longer timescales than DO events and most importantly it is obtained from an independent archive. Time series of these forcings are shown in Fig. 2c,d. The models' parameters are chosen such that the time-varying statistics are on average closest to those of the data. Specifically, by Monte Carlo simulation we generate many realizations for a fixed model parameter, compute time-varying statistics for each realization and then construct an average curve. Finally, the root-mean-square deviation (RMSD) from this curve with respect to the time-varying data statistic is computed. For best fit, we search for the least RMSD on a grid in parameter space. This corresponds to a numerical calculation of the maximum likelihood fit to the observed data. The two-process model is fitted to both statistics $E(t)$ and $P(t)$ simultaneously, by minimizing the normalized sum of the errors $\text{RMSD}_{E,P}$ to each of the statistics, defined as $\text{RMSD}_{sum} = \text{RMSD}_P/E_S + \text{RMSD}_P/P_S$.

The hypothesis tests are performed in the following way. For a given model we simulate a large number of realizations, which are collections of subsequent events with the same total duration as the record (104 kyr). For each realization we calculate the time-varying quantity of interest and the corresponding scalar test statistic. We then use the distribution of test statistics for a one-sided hypothesis test. The test simply counts how many test statistics in the ensemble are as large as or larger than the test statistic obtained from the data. Divided by the sample size, this yields a p-value, which estimates the probability of generating a random realization under the null hypothesis model that is at least as extreme than the observed data. We can reject the null hypothesis at a confidence level $\alpha$ if the p-value is smaller than $1 - \alpha$.

## 3 Results

The results of the hypothesis test on the stationary one- and two-process models are shown in Fig. 3a-c. The plots show distributions test statistics distributions of the respective null models and the corresponding test statistic value of the data. For the one-process model, the data test statistic lies well within the distribution, yielding a p-value of 0.16, as seen in Fig. 3a. Thus, we cannot rejected the null hypothesis at a level >85%. This indicates that the variations in the timing of warming transitions are consistent with a stationary Poisson process, i.e., without invoking variations in the rate parameter. Figures 3b,c show the hypothesis tests of the two-process model, yielding low p-values for both test statistics. The stationary two-process model is



thus rejected by the hypothesis tests with both test statistics at high confidence >98%.

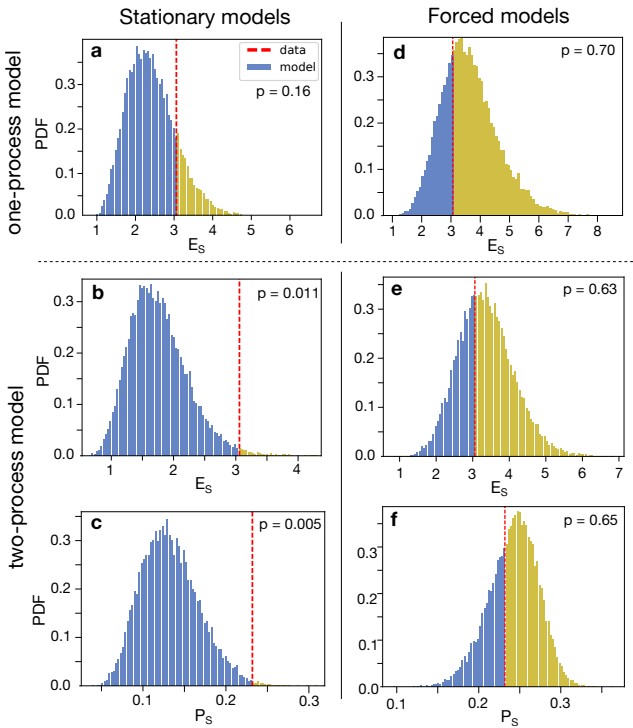

**Figure 3.** Empirical distributions from Monte Carlo simulation of the test statistics for the stationary (a)-(c) and the forced null model (d)-(f). The test statistics $E_S$ of the one-process models are shown in panels (a) and (d). The test statistics $E_S$ and $P_S$ of the two-process models are shown in panels (b) and (e), and (c) and (f), respectively. The position of the data test statistic within the distribution is marked in red and determines the p-value of the hypothesis tests.

To better visualize the outcomes of the hypothesis tests, we show confidence bands for the time-varying indicators from our Monte Carlo simulations in Fig. 4. The indicator $E(t)$ of the data lies within the 95% point-wise confidence band of the one-process model. Moreover, this band can be calculated analytically from the fact that the probability distribution of observing $k$ events in a time period $T$ is given by the Poisson distribution $P(k,T) = \frac{(\lambda T)^k}{k!}e^{-\lambda T}$. The cumulative distribution thereof allows us to calculate the probabilities of observing the minimal and maximal number of events per 20 kyr found in the data indicator $E(t)$. We find the probability to observe 2 or less events is $P = 0.047$ and to observe 12 or more events $P = 0.030$. This confirms that we cannot exclude the possibility of observing only 2 events and as much as 12 events during 20 kyr of the record at 95% confidence. The 95% confidence band of $E(t)$ for the two-process model in Fig. 4b is narrower and does not include the most extreme parts of the data curve. The same holds for the indicator $P(t)$, thus confirming that the two-process model can be ruled out with high confidence as null model for the observed sequence of events.





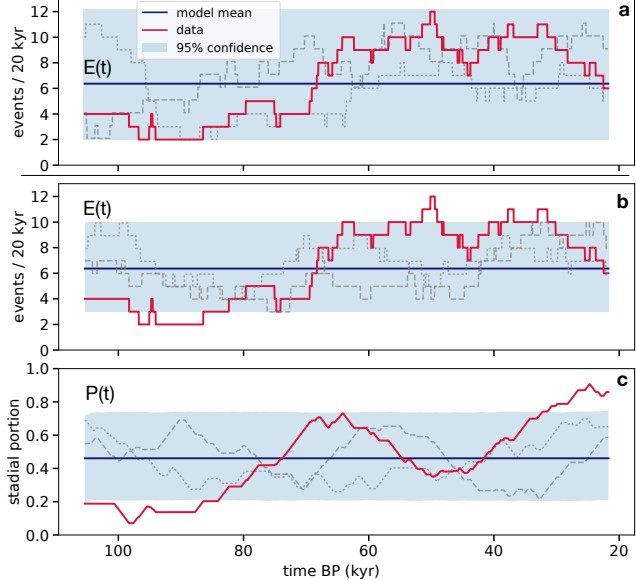

**Figure 4.** Point-wise 95% confidence bands and model mean (blue line) for the time-varying indicators $E(t)$ and $P(t)$ from Monte Carlo simulations. (a) $E(t)$ for the stationary one-process model. (b) and (c), $E(t)$ and $P(t)$ for the stationary two-process model. The indicators for the data are shown in red and two typical model realizations are shown in gray.

In the following we present the hypothesis tests performed on the one- and two-process models forced with insolation $S(t)$ and ice volume $I(t)$, which are both scaled to zero mean and range 1. Figure 5 shows the time dependent transition rates as obtained from the parameter fit. For the one-process model we obtain

$$\lambda(t) = 0.32 + 0.43 \cdot S(t) + 0.82 \cdot I(t). \tag{2}$$

The best-fit two-process model has stadial rate $\lambda_1(t)$ and interstadial rate $\lambda_2(t)$

$$\lambda_1(t) = 0.97 - 1.96 \cdot S(t) + 2.56 \cdot I(t)$$
$$\lambda_2(t) = 0.97 + 1.60 \cdot S(t) - 0.57 \cdot I(t). \tag{3}$$

The hypothesis tests for the fitted models are shown in Fig. 3d-f and yield high p-values, where the data statistic lies near the mode of the distributions. Note that we only measure the deviation of the time-varying statistics from a constant average

value. Thus, the statistical test is not targeted at evaluating the fit to the data, but merely at probing whether the fluctuations in time of the indicators are of the right magnitude. Goodness-of-fit can be seen by means of the confidence bands and mean of the time-varying indicators, as shown in Fig. 6. For both models, the mean indicators lie close to the data curves, which consequently lie within 95% confidence bands.

We additionally report how the goodness-of-fit of the forced models changes when using only partial forcing and thus a reduced number of parameters. When forcing the one-process model with both ice volume and insolation, we yield a RMSD





of the model mean $E(t)$ from the data curve of 1.42. Forcing with ice volume (insolation) only yields a best-fit RMSD of 1.64 (3.00). As baseline comparison, the RMSD from the unforced model to the data curve is equal to $E_S$, i.e. has a RMSD of 3.05. The model forced with ice volume fits the data only marginally worse than the model with both forcings and for comparison we show the mean time-varying indicator $E(t)$ for this model in Fig. 6a with a green dashed curve. For the two-process

model, we considered all combinations where both stadial and interstadial processes are only forced by either ice volume or insolation. Goodness-of-fit in the two-process model is given by the sum of errors of both indicators $\text{RMSD}_{sum}$. For the best-fit model with full forcing we find $\text{RMSD}_{sum} = 0.59$, whereas the baseline of an unforced model gives $\text{RMSD}_{sum} = 2.0$. Forcing both interstadial and stadial processes with ice volume (insolation) only yields a best-fit of $\text{RMSD}_{sum} = 0.90 \ (1.60)$. Using insolation for the stadials and ice volume forcing for the interstadials yields $\text{RMSD}_{sum} = 0.68$, while the converse choice

yields $\text{RMSD}_{sum} = 1.68$. Thus, the only reduced two-process model yielding a comparable goodness-of-fit compared to the model with full forcing is the model with stadial insolation and interstadial ice volume forcing. We show the mean time-varying indicators for this model in Fig. 6b,c with a green dashed curve.

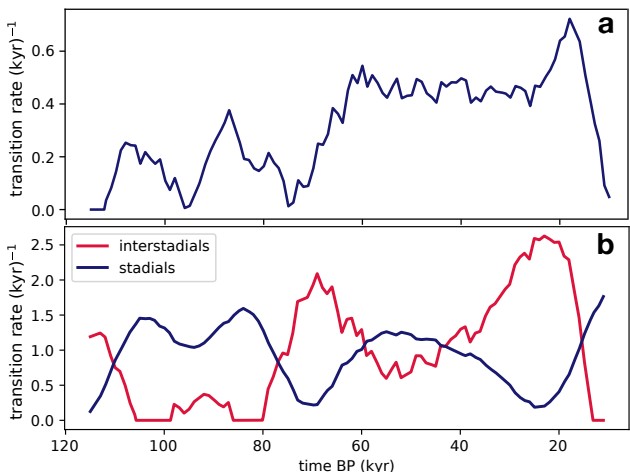

**Figure 5.** Time-varying transition rate parameters $\lambda(t)$ of the best-fit one-process (a) and two-process (b) models.

## 4   Discussion and Conclusions

Our first result considers only the warming events. While the distribution of waiting times in between warming events is well

modeled by an exponential distribution (not shown here), we show here that the number of events in a moving window of 20 kyr (and thus the mean waiting time) is clearly changing over time, but no more than would be expected from a realization of a stationary Poisson process. Thus, if there is a unique process giving rise to the warming transitions, it need not be changing over time due to external factors. Although the description of DO events solely by the timing of the abrupt warming is very simplistic, we still think is a useful result since the abrupt warmings are the most robust feature in ice core and other proxy





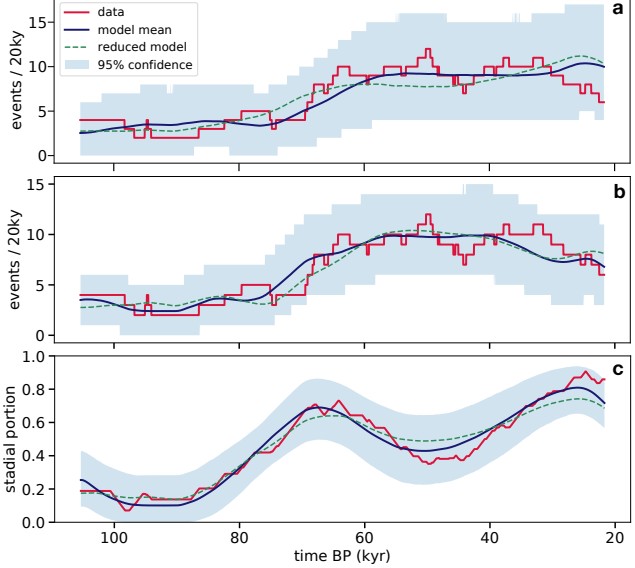

**Figure 6.** Point-wise 95% confidence bands and model mean (blue curve) for the time-varying indicators $E(t)$ and $P(t)$ from Monte Carlo simulations. (a) $E(t)$ for the best-fit non-stationary one-process model with full forcing. (b) and (c), $E(t)$ and $P(t)$ for the best-fit non-stationary two-process model with full forcing. The indicators for the data are shown in red. The green dashed line indicates the best-fit model mean curves for the respective models with reduced forcing, as described in the main text.

records and are commonly used to assess synchronicity and pacing of abrupt climate change in the last glacial.

The second result indicates, however, limits to the stationarity in the sequence of events as we increase the detail of description. Assuming two independent processes giving rise to transitions from stadials to interstadials and vice versa, the null
hypothesis of stationarity can be rejected with both our statistics. Specifically, both the variations in time of the number of warming events and the relative durations of stadials and interstadials are too large to be consistent with our two-process model using constant parameters. This model gives rise to a more regular sequence of warming events, compared to the one-process model. This is because one DO cycle is the sum of two independent processes and thus its duration is not distributed exponentially. In the limiting case of a DO cycle comprised of a very large number of independent processes, one finds a Gaussian
distribution of waiting times.

Next, we investigated improvements of the consistency of the models with the data by allowing their parameters to vary over time as a linear combination of two climate forcings. Choosing the best fit linear combination of forcings, we found the average time varying indicators of both models to match very well to the data curve. Thus, whereas the data was seen as a rather
out-lying realization that is consistent with a one-process model but not with a two-process model, when introducing forcings it becomes the expected behavior of the models. The goodness-of-fit follows from the correlation of the time-varying indicators





and the forcings, which can be seen in Fig. 2. For the ice volume proxy we find a Pearson correlation of $r^2 = 0.78$ with $P(t)$ and $r^2 = 0.58$ with $E(t)$. We assess significance of this correlation by fitting an AR(1) process to the linearly detrended ice volume and conduct a hypothesis test yielding a correlation with $P(t)$ of 0.33 with a p-value of $p = 0.035$ and thus significance at 95% confidence. In contrast, the correlation of the indicator $E(t)$ and ice volume does not go beyond the linear trend. We do

not assess the significance of correlations of insolation with the time-varying indicators, since it is difficult to find a good null model in this case.

Finally, we discuss the importance of ice volume and insolation in the best-fit one- and two-process models. In the one-process model, Eq. (2) shows that both increased insolation and ice volume lead to higher occurrence rates of events, with

the contribution of ice volume approximately twice as large as that of insolation. We note that the decrease in DO activity towards the Last Glacial Maximum is not captured by the model because the ice volume forcing is dominating. The individual contributions in the best-fit model do not completely capture the importance of the two forcings. There are directions in the likelihood landscape of parameters which are very flat. As a result, we found that the best-fit model with only ice volume forcing yields a fit only marginally worse than the best-fit model with both forcings. We thus conclude that ice volume is clearly the

more important control on the sequence of warming events. This is consistent with the findings of Mitsui and Crucifix (2017), where Bayesian model selection criteria show that global ice volume is a more important forcing than insolation in stochastic dynamical systems as models for Greenland ice core records.

In the two-process model the transition rates in stadials and interstadials are influenced by the forcing in opposite ways, as can be seen from Eq. (3): Transitions from stadials to interstadials become more likely for higher insolation and lower ice

volume, and vice versa for transitions from interstadials to stadials. For transitions from stadials, the contribution of ice volume is slightly larger than that of ice volume. The interstadial transition rate is dominated by insolation, which contributes three times more than insolation. With this model, the overall trend of mean waiting times in between warming events and of the stadial abundance is well captured, including the decrease in activity towards the LGM. Similar to the one-process model, we found a more parsimonious model which fits the data almost as well as the best-fit model with full forcing. This model uses

only insolation forcing for stadials and ice volume forcing for interstadials, which complements the analysis of the individual contributions in the fully forced model. We thus hypothesize that based on our study there is evidence for insolation control on average stadial duration and ice volume control on average interstadial duration. This finding could hint at two distinct mechanisms responsible for transitions in between regimes.

An exhaustive investigation of whether aforementioned finding and our model description is consistent with governing mechanisms for DO-type variability inferred from detailed data and realistic model studies is out of the scope of this paper. Nevertheless we conclude the discussion with some interpretations which are more speculative in nature. We begin with insolation control on stadial duration. Boreal summer insolation might influence the occurrence frequency of warming transitions by modulating the ice-ocean albedo feedback, which amplifies break-up or export of larger areas of sea ice. Sea ice decrease

could subsequently cause rapid warming through release of subsurface ocean heat (Dokken et al., 2013). Initial openings of





the sea ice cover might be created by wind stress. Evidence for stochastic wind stress forcing and subsequent sea ice changes have been reported in unforced model studies of rapid climate transitions (Drijfhout et al., 2013; Kleppin et al., 2015). To explain global ice volume control on interstadial duration we invoke different influences on the strength and stability of the interstadial (warm) mode of the Atlantic Meridional Overturning Circulation (AMOC). If we consider global ice volume as an

indicator of mean global climate, we find consistency with coupled climate simulations that show correlation of the stability of the warm AMOC branch to freshwater hosing and mean climate state (Kawamura et al., 2017). We furthermore note the study of Buizert and Schmittner (2015), where a correlation of individual interstadial duration and Antarctic temperatures from ice cores is established and explained by influences of Southern Ocean processes on the strength and stability of the AMOC. Given the strong similarity of the global ice volume record and Antarctic ice core records on longer time scales, this is closely

related to our findings. We finally note that in our model description, the trigger for warmings and coolings is stochastic and thus different from near-periodic DO cycles (Peltier and Vettoretti, 2014).

    In conclusion, we show that the long-term variations in DO warming event frequency, often described as millennial climate activity, is consistent with a memory-less stationary random process. From the data at hand we cannot exclude the possibility

that the long-term variations have occurred by chance. If we however divide a DO cycle into two independent processes governing warming and cooling, this is not true anymore and significant time-varying structure is detected. We thus propose a model that incorporates long-term variations through forcing of the parameters with external climate factors. We find good agreement with the data in a model where the mean duration of interstadial phases of the DO cycle are controlled by global ice volume and the stadial phases by boreal summer insolation. This finding can help to distinguish in between different

mechanisms that have been proposed to cause DO events.

*Competing interests.*   The authors declare no competing interests.

*Acknowledgements.*   This project has received funding from the European Union's Horizon 2020 research and innovation Programme under the Marie Sklodowska-Curie grant agreement No 643073.



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
