# Peer review of "Random and externally controlled occurrence of Dansgaard-Oeschger events"

_Climate of the Past, 2018_

## Referee Comment (RC1) · V. N. Livina (Referee) · 8 Mar 2018

The paper by Lohmann and Ditlevsen "Random and externally controlled occurrence of Dansgaard-Oeschger events" investigates possible mechanisms of the paleoclimate variability using several Poisson-process models and discusses the controls of warm and cold periods of the DO sequence in the NGRIP record. The paper is well-written and motivated, with detailed discussion of the models and their statistics, which are illustrated by clear figures. The paper is suitable for publication after a minor revision.

I think all the model parameters and statistics (RMSD, p-values) would be good to summarise in a table, which would help compare models at a glance.

If compare design of figures 2, 4, 6, notations P, E, I, S could be used in the same style, preferably in the label of the Y-axis rather than near the curves (sometimes with pointing lines sometimes without). Blue lines are better to replace by black ones, for better visibility (in my printout they look shaded).

Some text definitions could be accompanied by a suitable formula, see page 5, lines 13-14.

What is the authors' definition of "regularity" in page 4, line 12? Stationarity?

In page 2, line 19, it is better to say "consider" than "regard". The same in page 3, line 3.

In page 6, line 9, it is better to say "data source" than "archive".

In page 6, line 23: "as extreme as the observed data" (not "than")

In page 9, line 17: "it needs not"

In page 9, line 19: "we still think it is".

---

## Referee Comment (RC2) · T. Mitsui (Referee) · 16 Mar 2018

Dansgaard-Oeschger (DO) events are complicated phenomena, and their periodicity as well as their (non-)stationarity are still in debate. By using statistical hypothesis tests on simple event models, the authors test whether the sequence of DO events can be regarded as nonstationary or not. They conclude that we cannot reject the stationarity of the sequence if we focus only on warming events, but we can reject the stationarity if both warming and cooling events are taken into account. Furthermore, based on the model analysis, they propose different roles of external forcings on DO events such that warming events are mainly controlled by the global ice volume and the cooling events

by the boreal summer insolation.

This is a nice work which shows an external control of DO events, in a solid statistical way, with least assumptions. Their statistical hypothesis testing and model estimation look fine. The text is well written. The interesting hypothesis about different roles of external forcings on DO events is worthy to be reported. Thus, I recommend the publication of this article in Climate of the Past with the following minor revisions:

- In my first reading, I confused about the terms, "stadial rate" and "interstadial rate". I wondered if the stadail rate is the transition rate from stadial to interstadial or vice versa. I'm temped to call them "warming (cooling) rate" or " warming (cooling) event rate".

- Eq. (3) sounds counterintuitive because the insolation reduces the warming rate $\lambda_1$ and the ice volume increases the warming rate. Similarly, the insolation increases the cooling rate $\lambda_2$ and the ice volume decreases the cooling rate. Is there any possible explanation for this?

- Also I suggest to explicitly show the relation between the stadial rate and $S(t)$, and that between the interstadial rate and $I(t)$ for the reduced two-process model, like Eq. (3). Otherwise, it's not entirely clear whether the insolation (the global ice volume) indeed promotes or inhibits the warming (cooling) events.

- The integrated insolation above 350 $W/m^2$ (Huybers, 2006) is chosen as a forcing. Why don't you choose the summer solstice daily-mean insolation, which is also common? Is it a consequence of some optimization? If so, it is worthy to be mentioned.

- The authors mention "While the distribution of waiting times in between warming events is well modeled by an exponential distribution (not shown here)," (P9. Line 14-15). This is the fact from the observation since Ditlevsen's early works. The exponential distribution is true for the stationary one-process model but not true for the stationary two-process model as shown by by Eq. (1). The latter inconsistency is OK because the

authors rejects the model in the end. However, is the exponential distribution consistent with the non-stationary two process model? If so, why?

- How is the observation of the exponential distribution consistent with the following statement?: "In the limiting case of a DO cycle comprised of a very large number of independent processes, one finds a Gaussian distribution of waiting times" (P10. Line 9-10).

- In Eq. (1), both exponents are $-\lambda_1 T$. Is this right?

- P2. Line 12: Is "single events" fine?

- P3. Line 4 and in Fig. 6: "ky" -> "kyr" (if you want to correct)

- P3. Line 24-25: Svensson et al. (2006) -> (Svensson et al., 2006)

- P4. Line 1: "withing" -> "within"

- P8. Line 3: What do you mean by "range 1". Is this the value of the standard deviation?

---

## Author Comment (AC1) · 25 Mar 2018

We thank Valerie Livina for her thorough evaluation of our manuscript and helpful comments. In the following, we list all referee comments and our according responses.

1. "the model parameters and statistics (RMSD, p-values) would be good to summarise in a table"

We agree that this would be helpful as a summary. In the revised manuscript, we show a table of (1) model parameters, (2) hypothesis test results (p-values) and (3) Goodness-of-Fit of the model averages (RMSD) for both stationary and non-stationary

one- and two-process models. In order for the table not to become unmanageably big, we prefer to omit parameters and results for the reduced models, which are still given in the main text.

2. "figures 2, 4, 6, notations P, E, I, S could be used in the same style, preferably in the label of the Y-axis rather than near the curves"

We thank the referee for pointing out the different styles in the figures. We will adjust the figures in the manuscript.

3. "Blue lines are better to replace by black ones, for better visibility"

We agree and adjust the figures in cases where color is not needed for other purposes.

4. "text definitions could be accompanied by a suitable formula, see page 5, lines 13-14"

We included formulas for both E_S and P_S and hope that this increases readability.

5. "What is the authors' definition of "regularity" in page 4, line 12? Stationarity?"

We agree in the lack of clarity with the expressions "regularity"/"irregularity" and "non-stationarity". We used "regularity" as a colloquial term rather than a mathematical definition, denoting how variable the average event frequency (measured in a sliding window) is over time. We thus use the word to describe what our test statistic E_S measures. An equally spaced sequence of events is then considered fully regular. A strongly irregular event sequence in terms of E_S can be both due to non-stationarity or a very fat-tailed distribution of event waiting times. Thus it is not synonymous with "non-stationarity". Since we give a clear definition of E_S, we keep the colloquial use of "regularity"/"irregularity" in the manuscript and add a clear explanation of what we mean with "irregularity" in the introduction:

"In order to test the hypotheses, we regard the evolution of the number of warming events in a moving window of 20 kyr. This quantity measures how variable the average

event frequency is over time, a property which we denote as irregularity, and in the DO sequence it deviates strongly from a constant occurrence frequency of events over time."

6. "page 2, line 19, it is better to say "consider" than "regard". The same in page 3, line 3"

ok

7. "page 6, line 9, it is better to say "data source" than "archive" "

ok

8. "page 6, line 23: "as extreme as the observed data" (not "than")"

ok

9. "page 9, line 17: "it needs not" "

We would prefer the use of a modal auxiliary ("it need not be changing. . .") in this case. To make the sentence more clear we will change it to "it need not change . . . " in the manuscript.

10. "page 9, line 19: "we still think it is" "

ok

---

## Author Comment (AC2) · 25 Mar 2018

We thank Takahito Mitsui for carefully reading and evaluating of our manuscript. His comments have been very helpful and have improved the quality of the revised manuscript. In the following, we list all referee comments and our corresponding responses.

1. "I confused about the terms, "stadial rate" and "interstadial rate". I wondered if the stadail rate is the transition rate from stadial to interstadial or vice versa. I'm temped to call them "warming (cooling) rate" or " warming (cooling) event rate".

We agree that this terminology is confusing and will adopt a version of the reviewers proposition. Our original terminology was meant the following way: Stadial rate is the rate of transitioning from stadial to interstadial, thus it corresponds to the warming rate. In the same way, the interstadial rate can be referred to as the cooling rate. We would like to propose the following change: The transition rate from stadial to interstadial (and vice versa) is referred to as warming (cooling) transition rate.

2. "Eq. (3) sounds counterintuitive because the insolation reduces the warming rate $\lambda 1$ and the ice volume increases the warming rate. Similarly, the insolation increases the cooling rate $\lambda 2$ and the ice volume decreases the cooling rate. Is there any possible explanation for this?"

Thanks for this observation regarding Eq. (3). It is in fact a mistake, since we mixed up the right-hand sides for $\lambda 1$ and $\lambda 2$. When changing this, the interpretation is not counter-intuitive anymore: $\lambda 1$ is the warming transition rate (increase by insolation and decrease by ice volume) and $\lambda 2$ is the cooling transition rate (decrease by insolation and increase by ice volume). It has now been corrected.

3. "I suggest to explicitly show the relation between the stadial rate and S(t), and that between the interstadial rate and I(t) for the reduced two-process model, like Eq. (3). Otherwise, it's not entirely clear whether the insolation (the global ice volume) indeed promotes or inhibits the warming (cooling) events."

We agree that this will improve clarity and will add an equation in the manuscript.

4. "The integrated insolation above 350 W/m 2 (Huybers, 2006) is chosen as a forcing. Why don't you choose the summer solstice daily-mean insolation, which is also common? Is it a consequence of some optimization? If so, it is worthy to be mentioned.

We thank the referee for this important comment and agree that we need to address this in the manuscript. We did not use the integrated insolation forcing as a result of an optimization, but rather used it as a first choice since we believed it is the relevant

quantity for the phenomenon at hand, capturing the notion of positive degree days in high latitudes. However, we also conducted the fitting routine with daily-mean summer solstice insolation at 65 deg North, and obtained results that are equivalent to the case with integrated insolation. Specifically, we again find insolation control of stadial durations and ice volume control of interstadial durations. The fitting error with full forcing is then RMSD_sum = 0.62, and thus slightly worse than when using integrated insolation. We also find a reduced model with a very good fit of RMSD_sum = 0.69. The good agreement is not surprising because the two forcings look very much alike.

We thus added the following paragraph at the end of the discussion section:

"The results do not depend critically on the specific insolation forcing we used. To illustrate this, we also tried the daily-mean summer solstice insolation at 65 deg North and obtained results that are very much in line with what has been presented here. Specifically, we again find insolation control of stadial durations and ice volume control of interstadial durations. The fitting error with full forcing is RMSD_sum = 0.62, and thus only slightly worse than when using the integrated insolation presented in this paper. We also find a reduced model with a very good fit of RMSD_sum = 0.69. With this work we do not attempt to study which kind of insolation forcing might lead to the best fit, since the results would not be statistically significant given the small sample size of DO events and the fact that we already obtain a very close fit with both integrated and summer soltice insolation."

5. "The authors mention "While the distribution of waiting times in between warming events is well modeled by an exponential distribution (not shown here)," (P9. Line 14-15). This is the fact from the observation since Ditlevsen's early works. The exponential distribution is true for the stationary one-process model but not true for the stationary two-process model as shown by by Eq. (1). The latter inconsistency is OK because the authors rejects the model in the end. However, is the exponential distribution consistent with the non-stationary two process model? If so, why?"

We thank the referee for this question and would like to offer the following explanation. Instead of discussing whether (samples of) the distributions of the different models are consistent with one another (e.g. exponential distribution and non-stationary two-process model), we think it is more instructive to focus directly on whether the empirical data distribution is consistent with the different model distributions. The empirical distribution of inter-warming times in the data lies in fact very close to an exponential distribution of the same mean and is thus clearly consistent with it (using one-sided KS test: p=0.96). Interestingly, the data is also consistent with the distribution of the two-process model using parameters estimated from data, albeit with lower significance (p=0.30). When using the non-stationary two-process model, we find again strong correspondence of data and model distribution (p=0.92). The reason for this is following: The data distribution has a large dispersion (coefficient of variation CV = 1.129) , which is close to the one of the exponential distribution (CV=1.0). The stationary two-process model is, however, less dispersed (CV=0.708), as discussed in the manuscript. If we vary the parameters of the two-process model in time, the mean is varying and thus we expect the stationary distribution to be "smeared out" (i.e. more dispersed), which we indeed find for our best-fit non-stationary two-process model (CV=1.156). Thus it is close to the data distribution and presumably consistent with an exponential distribution. We would like to omit a discussion of this in the manuscript since the results of the paper are consistent with these considerations and the aim of this work was to go beyond the stationary statistics of waiting times in between warming events.

6. "How is the observation of the exponential distribution consistent with the following statement?: "In the limiting case of a DO cycle comprised of a very large number of independent processes, one finds a Gaussian distribution of waiting times" (P10. Line 9-10)."

Thanks for this interesting comment. We would like to point out that these two things are not supposed to be consistent, because we do not claim that a DO cycle is comprised of such a large sequence of independent processes. If we consider two independent processes, there is already a departure from exponential statistics. Here, we merely mention that it would tend to Gaussian statistics if there were more independent processes. The variance of this Gaussian would also decrease as the number of processes is increased. For clarity, we propose to address this by expanding the corresponding paragraph of the revised manuscript in the following way:

"This model gives rise to a more regular sequence of warming events, compared to the one-process model. This is because one DO cycle is the sum of two independent processes and thus its duration does not follow an exponential distribution (coefficient of variation CV = 1.0), but Eq. (1), which is less dispersed (CV = 0.708). In the limiting case of a DO cycle comprised of a very large sequence of N independent and stationary processes, one finds a Gaussian distribution of waiting times with decreasing variance as N grows. This would then correspond to an almost evenly-spaced sequence of events, which is not supported by the observations."

7. "In Eq. (1), both exponents are $-\lambda_1 T$. Is this right?"

Thanks. The first exponent is $-\lambda_2 T$, which has been corrected in the manuscript.

8. "P2. Line 12: Is "single events" fine?"

Changed it to "individual events".

9. "P3. Line 4 and in Fig. 6: "ky" -> "kyr" (if you want to correct)"

ok

10. "P3. Line 24-25: Svensson et al. (2006) -> (Svensson et al., 2006)"

ok

11. "P4. Line 1: "withing" -> "within" "

ok

12. " P8. Line 3: What do you mean by "range 1". Is this the value of the standard

deviation? "

With "range 1" we mean that the amplitude (maximum – minimum value) of the signal is 1. Has been clarified in the manuscript.